# Low-Intensity, High-Frequency Grazing Strategy Increases Herbage Production and Beef Cattle Performance on Sorghum Pastures

**DOI:** 10.3390/ani12010013

**Published:** 2021-12-22

**Authors:** Thales Baggio Portugal, Leonardo Silvestri Szymczak, Anibal de Moraes, Lidiane Fonseca, Jean Carlos Mezzalira, Jean Víctor Savian, Angel Sánchez Zubieta, Carolina Bremm, Paulo César de Faccio Carvalho, Alda Lúcia Gomes Monteiro

**Affiliations:** 1Department of Crop Production and Protection, Federal University of Paraná, Curitiba 80035-050, Brazil; sisz.leonardo@gmail.com (L.S.S.); anibaldemoraes@gmail.com (A.d.M.); aldaufpr@gmail.com (A.L.G.M.); 2CONSIPA—Consulting on Integrated Crop-Livestock Systems, Ponta Grossa 84015-500, Brazil; lidianefnc@gmail.com (L.F.); mezzalirajc@gmail.com (J.C.M.); 3Grazing Ecology Research Group, Federal University of Rio Grande do Sul, Porto Alegre 91540-000, Brazil; gello_zuva@hotmail.com (A.S.Z.); carolina.bremm@yahoo.com.br (C.B.); paulocfc@ufrgs.br (P.C.d.F.C.); 4Programa Pasturas y Forrajes, Instituto Nacional de Investigación Agropecuaria (INIA), Estación Experimental INIA Treinta y Tres, Ruta 8 km 281, Treinta y Tres 33000, Uruguay; jvsavian@gmail.com; 5Sheep and Goat Production and Research Center, Federal University of Parana, Curitiba 80035-050, Brazil

**Keywords:** optimal sward structure, grazing management, rotational stocking, sward defoliation

## Abstract

**Simple Summary:**

Finding smart management targets to improve livestock production and make it sustainable are very important for livestock in the tropics. We assessed the effects of high-intensity and low-frequency (HILF) vs. low-intensity and high-frequency (LIHF) grazing on herbage production and performance of beef cattle grazing sorghum pastures. The LIHF resulted in shorter rest periods when compared with the HILF. The greater leaf lamina mass in LIHF allowed greater sward light interception at post-grazing, resulting in greater total herbage production than HILF. The average daily gain (ADG) was greater for the LIHF than for the HILF treatment; however, even with a greater stocking rate in the HILF, there was no difference for LW gain per ha. Our findings demonstrated that the LIHF strategy that is based on offering to the animals an optimal sward structure to favor the herbage intake rate fosters greater herbage production, harvesting efficiency, and ADG without compromising LW gain per area, despite the lower herbage harvested per stocking cycle. Therefore, we conclude that the classic trade-off between animal performance and forage production could be offset on tropical grasses grazed by beef cattle only by adjusting grazing management according to a LIHF grazing management strategy.

**Abstract:**

We assessed the effects of high-intensity and low-frequency (HILF) vs. low-intensity and high-frequency (LIHF) grazing on herbage production and performance of beef cattle grazing sorghum pastures. The experimental design was a complete randomized block with two treatments and four replicates (paddocks), carried out in 2014/15. The management target of 50 and 30 cm for pre- and post-grazing, respectively, a LIHF grazing management strategy oriented to maximize beef cattle herbage intake per unit time, was compared with a HILF grazing management strategy of 80 and 20 cm for pre- and post-grazing, respectively, aiming to maximize herbage accumulation and harvest efficiency. Sixteen Brangus steers of 15-month-old and 265 ± 21 kg of live weight (LW) were randomly distributed to paddocks (experimental units). The LIHF resulted in shorter rest periods when compared with the HILF. The greater leaf lamina mass in LIHF allowed greater sward light interception at post-grazing, resulting in greater total herbage production than HILF (7581 and 4154 kg DM/ha, respectively). The average daily gain (ADG) of steers was greater for the LIHF than for the HILF treatment (0.950 and 0.702 kg/animal, respectively); however, even with a greater stocking rate in the HILF, there was no difference for LW gain per ha, with an average of 4 kg LW/ha/day. Our findings demonstrated that the LIHF strategy that is based on offering to the animals an optimal sward structure to favor the maximum herbage intake rate fosters greater herbage production, harvesting efficiency, and ADG without compromising LW gain per area of beef steers, despite the lower herbage harvested per stocking cycle.

## 1. Introduction

Grazing management strategies [1,2] affect herbage growth [3], animal performance [4,5], and the functioning of the pastoral ecosystem [6]. In rotational stocking, grazing strategies are designed by the control of the intensity and frequency of animal defoliation on spatiotemporal scales [7]. Intensity/frequency of grazing are the same in continuous stocking, but they are not directly controlled by the manager, as they are associated with animal decisions [8].

The aim in controlling defoliation by rotational stocking strategies originates from the desire to determine what, and to what extent, animals should graze (intensity), and to control plant recovery to defoliation (frequency). In general, maximum herbage accumulation is set as the optimal time to start grazing [4,9,10], while intense grazing (i.e., low residual sward height) is usually imposed as the limit of sward depletion [11,12,13]. These criteria of grazing management result in high-intensity, low-frequency (HILF) defoliation, and aim to increase the instantaneous herbage harvest efficiency, herd dry matter (DM) intake, and output per unit area [14,15].

The HILF grazing strategies are widely adopted in rotational stocking [1]. They are the basic criteria of numerous known fenced systems such as Holistic grazing management, short-duration grazing, time-controlled grazing, and cell grazing (see di Viglizo et al. [16]). Conversely, low-intensity indicates that grazing animals may select according to their preference, at least for certain plant morphological components (e.g., leaves), taking the best and leaving the rest [17], and high-frequency would not allow the required time to recover after defoliation (see Schons et al. [18]). Overall, LIHF is a theoretical grazing management strategy that does not fit the rotational stocking main original premises.

In traditional rotational stocking, high grazing intensity, mainly achieved by increasing stocking rate, is imposed to full exploitation of the pasture area for maximum farm profit (see Fariña and Chilibroste [19]). Conversely, Carvalho [17] proposed a grazing management concept, named ‘Rotatinuous’ stocking, that sets the limits of pre- and post-grazing sward heights that allow animals to achieve maximum forage intake rate and sustain it at any time while grazing. Numerous studies have defined these limits for several temperate and tropical grass species (e.g., [20,21,22,23,24]), resulting in sward height targets to design LIHF grazing strategies. Schons et al. [18] applied this criterion on Italian ryegrass (*Lolium multiflorum* Lam.) and observed the optimization of both primary and secondary production per hectare, compared with a traditional HILF rotational stocking. This first evidence with temperate pastures challenged the LIHF rotational grazing strategies commonly unutilized. 

Tropical forages are not easy to manage, as they display a high herbage mass accumulation rate, and their quality usually deteriorates rapidly. Consequently, HILF grazing strategies using high stocking densities are usually preferred in the tropics, as lenient grazing with fast-growing forages could lead to stem forage structures, providing smaller bite mass and herbage intake (see Benvenutti et al. [25]). We challenged this assumption by investigating the effects of imposing the LIHF strategy on fast-growing annual C4 grasses, such as sorghum pastures. This species is of great importance on farms due to its high forage production potential, high nutritional value, and drought tolerance (Soares et al. [26]). However, it presents stem vegetative tillers elongation and sward structure change in a vegetative stage, making management difficult [27]. Considering this, we hypothesized that a lenient but frequent grazing strategy called ‘Rotatinuous’ stocking applied on sorghum pastures could optimize both herbage and animal production. To test this hypothesis, we evaluated the primary and secondary production of beef cattle managed under rotational stocking with contrasted HILF and LIHF strategies. 

## 2. Material and Methods

### 2.1. Experimental Area

The experiment was conducted at the Agronomic Experimental Station (AEE) of the Federal University of Rio Grande do Sul (UFRGS), in Eldorado do Sul city, southern Brazil (latitude 30°05′ S, longitude 51°39′ W, and altitude 46 m). The climate is subtropical with a warm humid summer (Cfa) according to the Köeppen classification [28], with an average annual temperature of 18.8 °C and average annual precipitation of 1455 mm.

The soil at the experimental site was classified as a Plinthosol [29]. Soil chemical analyses (depth 0–20 cm) indicated 2.29% of organic matter ((C organic × 1.74)/10), pH of 4.12, exchangeable aluminum of 0.55 cmol_c_/dm^3^, K of 0.27 cmol_c_/dm^3^, Ca of 1.87 cmol_c_/dm^3^, Mg of 0.82 cmol_c_/dm^3^, base saturation of 38.5%, and P of 26.65 mg/dm^3^. During the experimental period, from November 2014 to March 2015, the mean air temperature and total rainfall were 23.3 °C and 608 mm, respectively (AEE-UFRGS).

Four hectares of sorghum (mixed *Sorghum bicolor* (L.) Moench and *Sorghum sudanense* (Piper) Stapf cv. EMBRAPA BRS 802) [30] were sown on 4 November 2014. No-tillage was realized with 18 kg seeds per ha, and 0.22 m spacing between rows (sown in crossed lines, perpendicular to each other). Before sowing, the Italian ryegrass in the experimental area was desiccated with Isopropylamine salt of Glyphosate at a 3 L/ha dosage. At the time of sowing, 40 kg *n*/ha, 240 kg P_2_O_5_/ha and 120 kg K_2_O/ha were applied. Additionally, urea was applied on 2 December 2014 (150 kg n/ha) and on 11 February 2015 (50 kg n/ha).

### 2.2. Experimental Design and Pasture Management

The experiment was designed as complete randomized blocks, with two treatments and four replicates (*n* = 8 paddocks or experimental units). Two grazing management strategies were tested: a low-intensity, high-frequency grazing strategy (LIHF) named ‘Rotatinuous’ stocking, with pre- and post-grazing sward heights of 50 and 30 cm, respectively, aiming to optimize and sustain the beef cattle herbage intake rate at any time while grazing sorghum pastures [22], and the second treatment was a high-intensity, low-frequency grazing strategy (HILF) depicting the traditional rotational stocking, with pre-grazing of 80 cm and 20 cm at post-grazing [31], which targets optimum forage mass accumulation and maximum instantaneous herbage harvesting.

The stocking season was from 6 December 2014 to 3 March 2015. The experimental area was divided into eight paddocks of 0.5 ha each. A one-day occupation period was adopted on the strip for both treatments. The rest period varied according to plant growing conditions and represented the time (days) elapsed in the regrowth to recover from post- to pre-grazing sward height targets. The stocking cycle was defined by the time of the occupation period plus the rest period [32]. Animals were moved daily from the strip between 16 and 17 h. The number of strips for each treatment was flexible and a consequence of targeted pre-grazing sward heights and plant growth.

### 2.3. Sward Measurements

The pre- and post-grazing sward heights were measured every two days taking 100 readings on each strip with a sward stick [33]. Additionally, at the beginning and end of each stocking cycle, the pre- and post-grazing herbage masses were assessed by random allocation of quadrants (0.405 m^2^) at four points within the strip, and clipping the herbage at ground level.

Daily herbage accumulation rate (kg DM/ha/day) was calculated by the difference between the pre-grazing herbage mass of the following cycle subtracted from the post-grazing mass of the previous one and divided by the number of days elapsed. All herbage samples were dried in a forced-air oven at 55 °C for 72 h. Total herbage production (kg DM/ha) was obtained by the sum of herbage mass at the beginning of the stocking season and the herbage accumulation rate multiplied by the number of grazing days of the entire stocking season. At each herbage clipping, leaf lamina, pseudostem, stem plus sheath, dead material, and inflorescence were separated and dried in a forced-air oven at 55 °C for 72 h to estimate the morphological components (leaf:stem ratio).

The light interception was measured using a Decagon AccuPAP LP-80^®^ Ceptometer, by the difference between the top of the sward and at the soil surface, during the whole stocking season. The measurements were performed between 11 and 13 h, only when conditions (full sunlight) permitted, roughly every nine days. To minimize the plant distribution effect in the area, ten measurements were taken north-southwards and another ten east-westwards in each strip (pre- and post-grazing).

Tiller counting was performed using a 0.405 m^2^ rectangle at four random points within the strip (always at pre-grazing); this was always performed on the same strip at the beginning of each stocking cycle. Yet, in the LIHF it started from the fourth cycle onwards, totaling 12 measurements, whereas, in the HILF treatment, it started from the second stocking cycle onwards and 15 days (strips) later in order to increase the number of samples.

### 2.4. Animal Measurements

Sixteen Brangus steers aged 15 ± 1 months and 265 kg ± 21 kg (mean ± s.d.) live weight (LW) were randomly allocated into paddocks, that is, two test-steers per paddock. Additional non-experimental similar steers were used according to the put-and-take technique [34] to maintain the targeted sward heights.

Before the commencement of the trial, all animals were treated with an anthelmintic, and the control of external parasites such as ticks and horn-flies was performed when necessary [35,36].

Every 21 days, the animals were fasted for 12 h and afterwards weighed for LW gain measurement. Average daily gain (ADG, kg/animal) was calculated by dividing the LW change of test animals between two consecutive weighing events by the number of days elapsed.

The stocking rate (kg LW/ha) in each stocking cycle was calculated according to the following equation:(1)$$Stocking\;rate\;(kg\;LW/ha)=\frac{\left(\frac{LW\times A}{A_{sg}}\right)}{D}$$
where *LW* is the live weight of the test and put-and-take animals, *A* is the area of a hectare, *A_sg_* is the strip-grazing area (m^2^) and *D* is the number of days of the stocking cycle. Daily *LW* gain per area (*kg LW/ha*) was obtained by multiplying the stocking rate (expressed in the number of animals per hectare) by the ADG of the test animals.

Harvested herbage mass per stocking cycle (*kg DM/ha*) was the difference between pre- and post-grazing herbage mass measurements in a strip-grazing. Total herbage mass harvested (*kg DM/ha*) during the stocking season was calculated by the sum of harvested herbage mass in all stocking cycles.

### 2.5. Statistical Analysis

Data were submitted to analysis of variance (ANOVA) at a 5% significance level (*p* < 0.05). The assumptions of normality (Shapiro-Wilk, *p* > 0.05), homogeneity of variance (Bartlett, *p >* 0.05), and independence of residuals (visual analysis) were checked. The statistical model for the analysis of sward variables measured in the pre- and post-grazing time included the fixed effects of grazing management strategy, time, and their interaction. Stocking cycle and block were considered random effects. For the variables measured by stocking cycle, grazing management strategy was included in the model as a fixed effect and stocking cycle and block as a random effect. The model for variables obtained by stocking season included grazing management strategy as a fixed effect and block as a random effect. The analyses were performed with the R statistical software version 4.0.2 [37]. The *lme4* package [38] was used for analyzing the statistical models; when significant differences were detected, means were compared by Student’s t-test (*p* < 0.05) using the *emmeans* package [39].

## 3. Results

Table 1 shows the structural characteristics of sorghum pastures. The sward heights were close to the proposed targets, with pre-grazing average sward heights of 47.5 cm for LIHF and 83.6 cm for HILF, and post-grazing average sward heights of 33.7 and 28.3 cm for LIHF and HILF, respectively (*p* < 0.001). This represents an average sward height depletion of 29% and 66% for LIHF and HILF treatment, respectively (*p* < 0.001).

The HILF treatment showed greater herbage mass, leaf lamina mass, and stem mass at the pre-grazing (*p <* 0.01) when compared with the LIHF treatment. In the post-grazing, LIHF treatment presented greater leaf lamina mass (*p <* 0.001), while herbage mass and stem mass were similar between grazing management strategies (*p* > 0.05). Pre-grazing leaf-to-stem ratio and sward light interception were greater for the HILF (*p* < 0.001); nevertheless, these variables were greater for the LIHF in post-grazing (*p* < 0.001). Tiller density did not differ between treatments (*p* = 0.155), with an average of 43.5 tillers/m^2^.

The grazing management strategy affected the number of stocking cycles and rest periods (*p* < 0.001; Table 2). The rest periods of LIHF were of 5 days, performing 15 stocking cycles, and the HILF completed rest periods within 31 days, performing three stocking cycles.

The daily herbage accumulation rate was greater in the LIHF (*p* = 0.010) compared with the HILF treatment, with an average of 138 and 88 kg DM/ha/day, respectively. Accordingly, total herbage production was also greater (*p* = 0.004) in the LIHF than in the HILF, with an average of 11,639 and 5911 kg DM/ha, respectively. The herbage harvested by the animals per stocking cycle was greater in the HILF treatment (*p* < 0.001); however, the total herbage harvested per stocking season was greater in the LIHF than in the HILF treatment (*p =* 0.001; Table 2), with an average of 7581 and 4154 kg DM/ha, respectively.

Table 3 shows animal responses as affected by grazing management strategies. The ADG was greater (*p* = 0.017) in the LIHF than in the HILF treatment, with an average of 0.950 and 0.702 kg/animal, respectively. Differently, the stocking rate was greater (*p* < 0.001) in the HILF than in the LIHF treatment (1756 and 1370 kg LW, respectively). However, LW gain per hectare remained unaffected (*p* = 0.950), with an average of 4 kg LW/ha/day.

## 4. Discussion

Several reports suggest that there exists an incompatibility of maximizing both individual animal output and full exploitation of the area [15,40]. Carvalho [17] refers to this as the apparent trade-off of pasture management. However, previously, with temperate pastures [18,41] and in this work, with tropical forages, it is demonstrated that this classic trade-off can be overcome by adopting a LIHF, based upon animal-behavioral response as a function of sward structure (i.e., intake rate).

We found a difference of 5725 kg DM/ha for total herbage production to LIHF compared with HILF. These results may be related to the higher daily herbage accumulation rate in LIHF, i.e., 138 and 88 kg DM/ha/day for LIHF and HILF, respectively (Table 2).

In the HILF treatment, i.e., high grazing intensity, there were profound effects on herbage growth because it drastically reduced the residual leaf mass during grazing down (Table 1), thus changing the dynamics between acquisition, use, and carbon stock of the plant [4,42]. Conversely, the greater herbage accumulation in LIHF can be explained by the narrow range of sward height depletion, resulting in short stocking cycles (Table 2), in turn explained by the greater post-grazing leaf lamina mass, which resulted in greater light interception (Table 1) and photosynthetic capacity of swards [42,43,44,45]. Other studies also support these responses. For example, Schons et al. [18], Szymczak [46], and Gomide et al. [47] found an increase in the productive potential of Italian ryegrass, tall fescue (*Schedonorus arundinaceus*), and Aries Guinea grass (*Megathyrsus maximum*), respectively, under conditions of low defoliation intensity.

Apart from affecting herbage growth, it was also observed that high instantaneous harvest levels through severe grazing compromised the total herbage harvesting at the end of the stocking season (Table 1). Sbrissia et al. [48] observed this response with kikuyu grass (*Pennisetum clandestinum*), showing that as long as the sward depletion does not surpass 50%, the herbage growth rate does not vary among pre-grazing sward heights going from 15 to 25 cm and that a similar response would be expected with sward depletion of 40% as in the LIHF treatment criteria. In studies with *Pennisetum purpureum* [49] and *Panicum maximum* [50], severe defoliation is particularly undesired, as it reduced tussock cover; the inverse occurred, without stem elongation, when the depletion was 50% of the pre-grazing sward heights determined by the criteria of 95% of light interception [4].

Tillers are sensitive to the amount and quality of light intercepted [42,51]. In rotational stocking, the greater light entering the sward [52] after severe grazing promotes tillering and leafy growth [12]. These processes would have been expected to be associated with HILF treatment, in which around 66% of the pre-grazing sward height was depleted, as demonstrated in this experiment (Table 2).

Several studies have reported the relationship between grazing and tillering (e.g., Matthew et al. [53]; Nelson [54]; Da Silva and Nascimento Junior, [55]). Sward defoliation affects the auto-adaptive property of plants by controlling light interception and quality inside plant sward, providing the reduction of the leaf area index and consequent light tillering signaling [56,57,58]. However, we found no significant effect of the grazing management strategy on the average number of tillers (Table 1). Therefore, we attribute this response to another genetic characteristic intrinsic to sorghum pastures. Kebrom and Mullet [59] reported increased dormancy and decreased growth of buds responsible for tillering in sorghum plants soon after defoliation. To Kong et al. [60] and Alam et al. [61], sorghum has a great genetic influence on tillering regulation, suggesting that there is a predominance of alleles responsible for apical dominance. Yet, Tamele [62] showed a decrease in the number of tillers in sorghum pastures after the second stocking cycle, an intrinsic characteristic of this species.

Severe sward defoliation results in lower individual herbage intake [63,64], and thus in suboptimal individual animal performance [18], but also in greater instantaneous herbage intake per area [40,65]. This was demonstrated in the HILF treatment, as a greater stocking rate was mandatory to reach full exploitation of the area [15,40] and optimize farm profit [19]. In this case, this trait is represented by low post-grazing sward heights, and, hence, lower herbage allowance and individual herbage intake, which can explain the lower ADG and also the greater herbage harvesting per cycle (Table 3). This creates the instantaneous perception of no herbage waste, as most of the herbage on offer is harvested. Nevertheless, as noticed, this reduces herbage growth and extends stocking cycles when compared with the LIHF treatment. With this, lower herbage harvesting per cycle and greater herbage accumulation rate result in many more stocking cycles (15 vs. 3 for LIHF and HILF, respectively) and in two times more herbage harvested at the end of the stocking season (Table 2). With this grazing management strategy, instantaneous perception of waste was accepted for the sake of maximized primary and secondary production at the end of the stocking season. Indeed, the amount of herbage harvested (7581 vs. 4154 kg DM/ha) in relation to that produced (11,639 vs. 5911 kg DM/ha) was 65% and 70% for LIHF and HILF, respectively; thus, maximizing instantaneous herbage harvest efficiency reduces the overall amount of herbage harvested in the long-term. However, the herbage mass not harvested by the animals in each strip will remain to the next stocking cycle to be grazed, mainly because the rest period is short (high grazing frequency). Thus, the greater herbage production in the LIHF treatment can contribute to nutrient recycling and soil conservation in the long-term. For instance, Assmann et al. [66] showed that heavy grazing intensity results in losses of carbon and nitrogen and soil organic matter degradation when compared with moderate and light grazing intensities.

Increasing the intake of a good quality herbage is necessary to optimize LW gain of grazing animals [67]. We can assume that the herbage ingested by animals in the LIHF grazing strategy presented greater nutritive value, mainly because the animals ate the top stratum of the sward composed mainly by leaves, and were not induced to eat the bottom parts of plants, as the animals under the HILF grazing strategy were forced. This response was previously demonstrated with sheep grazing Italian ryegrass pastures managed under LIHF and HILF. That is, the nutritive value of herbage was greater for the LIHF, which means lower contents of neutral detergent fiber, acid detergent fiber, lignin, and greater content of crude protein and organic matter digestibility in each stocking period (strip) [68,69] and over the stocking season [63].

Despite the greater ADG and lower stocking rate in LIHF treatment, and thus lower instantaneous herbage harvest efficiency, the daily LW gain per hectare was not affected by grazing management (Table 3). This overcomes the classic trade-off commonly reported in grazing ecosystems, showing that secondary production is not reduced with grazing management optimizing individual LW gain through lenient grazing. Despite treatments displaying similar daily LW gain per hectare, the lower growth rate of HILF animals could impair some economic and environmental benefits. For instance, lower individual performance can extend the time to slaughter [70] and carcass quality [41,71], which in turn could reduce the competitiveness of the beef supply chain [72,73]. Additionally, as proved by Schons et al. [18] with temperate pastures grazed by sheep, we confirmed with tropical pastures grazed by beef cattle that a high stocking rate is not synonymous with high animal production, that is, a moderate grazing intensity based on animal responses (intake rate) such as ‘Rotatinuous’ stocking can be used for different pastures and systems to improve ruminant livestock production. Furthermore, lower animal performance increases fattening-to-slaughter CH_4_ emission from enteric fermentation [74,75,76] and results in animal commodities with greater carbon footprint [64,68,74].

## 5. Conclusions

Our findings highlighted that when sorghum pastures are managed under rotational stocking with pre- and post-grazing sward height targets of 50 and 30 cm, respectively (i.e., LIHF), there is a greater herbage accumulation rate and, at the end of the stocking season, more herbage is produced and harvested by the animals. In addition, in the LIHF treatment, even with a lower stocking rate, animals gained more LW individually (ADG) without impairing daily LW gain per hectare. Finally, we demonstrated that the classic trade-off between animal performance and forage production could be offset on tropical grasses grazed by beef cattle only by adjusting grazing management according to a LIHF (‘Rotatinuous’ stocking) grazing management strategy.

## Figures and Tables

**Table 1 animals-12-00013-t001:** Characteristics of sorghum pastures grazed by beef cattle under different grazing management strategies (LIHF and HILF).

Variables	LIHF	HILF	SEM	*P* _S_	*P* _T_	*P* _S × T_
Sward height (cm)						
Pre-grazing	47.5 Ab	83.6 Aa	1.75	<0.001	<0.001	<0.001
Post-grazing	33.7 Ba	28.3 Bb	0.38			
Herbage mass (kg DM/ha)						
Pre-grazing	1565 Ab	2390 Aa	81.9	0.003	<0.001	<0.001
Post-grazing	1144 Ba	1000 Ba	44.8			
Leaf lamina mass (kg DM/ha)						
Pre-grazing	492 Ab	878 Aa	41.7	0.002	<0.001	<0.001
Post-grazing	274 Ba	29 Bb	21.4			
Stem mass (kg DM/ha)						
Pre-grazing	971 Ab	1324 Aa	50.4	0.001	<0.001	0.006
Post-grazing	786 Ba	846 Ba	36.4			
	Leaf:stem ratio					
Pre-grazing	0.52 Ab	0.74 Aa	0.04	<0.001	<0.001	<0.001
Post-grazing	0.37 Ba	0.04 Bb	0.03			
Light interception (%)						
Pre-grazing	35.4 Ab	59.4 Aa	2.2	0.019	<0.001	<0.001
Post-grazing	22.5 Ba	8.5 Bb	1.5			
Tiller density (tillers/m^2^)	46.1	40.9	1.7	0.155	-	-

LIHF = low-intensity and high-frequency grazing strategy; HILF = high-intensity and low-frequency grazing strategy; DM = dry matter; SEM = standard error of the mean. *P*_S_; *P*_T_; *P*_S × T_ correspond to *p*-values, where S = grazing management strategy (LIHF and HILF), T = time (pre- and post-grazing), and S×T = interaction between grazing management strategy and time factors. Distinct letters, uppercase in column and lowercase in line, differ by Student’s *t*-test (*p* < 0.05).

**Table 2 animals-12-00013-t002:** Stocking cycles, herbage production, and total herbage harvested by beef cattle under different grazing management strategies (LIHF and HILF).

Variables	LIHF	HILF	*p*-Value	SEM
Number of stocking cycles	15	3	<0.001	0.4
Rest period (days)	5	31	<0.001	0.2
Daily herbage accumulation rate (kg DM/ha)	138	88	0.010	11.5
Total herbage production (kg DM/ha)	11,636	5911	0.004	1167
Herbage harvested per stocking cycle (kg DM/ha)	426	1351	<0.001	78.4
Total herbage harvested (kg DM/ha)	7581	4154	0.001	712

LIHF = low-intensity and high-frequency grazing strategy; HILF = high-intensity and low-frequency grazing strategy; DM = dry matter; SEM = standard error of the mean.

**Table 3 animals-12-00013-t003:** Stocking rate and performance of beef cattle grazing sorghum pastures under different grazing management strategies (LIHF and HILF).

Variables	LIHF	HILF	*p*-Value	SEM
Stocking rate (kg LW/ha)	1370	1756	<0.001	74.9
Average daily gain (kg/animal)	0.950	0.702	0.017	0.06
Daily LW gain (kg/ha)	4.0	4.0	0.950	0.16

LIHF = low-intensity and high-frequency grazing strategy; HILF = high-intensity and low-frequency grazing strategy; LW = live weight; SEM = standard error of the mean.

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
