# Peer review of "Low-Intensity, High-Frequency Grazing Strategy Increases Herbage Production and Beef Cattle Performance on Sorghum Pastures"

_animals, 2021, doi:10.3390/ani12010013_

Round 1
Reviewer 1 Report
Lines 143-146: could you please describe closely how you dried the cut grass? (°C / hours, oven); also in lines 149/150: describe it a bit closer.
Line 158: I did not understand that clearly: was it done just in one strip? So you should not write "per strip", but "always in the same strip" on line 149.
Chapter 2.5: did you observe any animal health parameters, especially parasites? Please write a sentence about observation of animal health. Please write some sentences on animal health, also in the results part.
After Line 235: Could you please also mention at what age (days) the animals were slaughtered in both systems; at least the ones that were not added to the system HILF: in the discussion part you mention that it is more efficient (and also has an impact on meat qualtity and climate) to fatten animals faster, so it would be interesting to see this difference here, in the results part.
Chapter 4.: Discussion: I think your work is very interesting and important and I also like, how you discuss it. But I thought that some sentences on fertilizing (animal manure / plant residuals) and soil compaction should be added. Do you expect them to be about the same in both systems or are there differences or do you think such differences could be observed in further studies or on the long term? These subjects should at least be mentioned in the discussion part.
Author Response
- Lines 143-146: could you please describe closely how you dried the cut grass? (°C / hours, oven); also in lines 149/150: describe it a bit closer.
R: We have added the suggested information.
Lines 158-166: Daily herbage accumulation rate (kg DM/ha/day) was calculated by the difference between the pre-grazing herbage mass of the following cycle subtracted from the post-grazing mass of the previous one, and divided by the number of days elapsed. All herbage samples were dried in a forced-air oven at 55°C for 72 hours. Total herbage production (kg DM/ha) was obtained by the sum of herbage mass at the beginning of the stocking season and the herbage accumulation rate multiplied by the number of grazing days of the entire stocking season. At each herbage clipping, leaf lamina, pseudostem, stem plus sheath, dead material and inflorescence, were separated and dried in a forced-air oven at 55°C for 72 hours to estimate the morphological components (leaf:stem ratio).
- Line 158: I did not understand that clearly: was it done just in one strip? So you should not write "per strip", but "always in the same strip" on line 149.
R: Ok. We modified the phrase to be more consistent.
Line 173-175: Tiller counting was performed using a 0.405 m2 rectangle at four random points within the strip (always at pre-grazing); this was performed always on the same strip, at the beginning of each stocking cycle.
- Chapter 2.5: did you observe any animal health parameters, especially parasites? Please write a sentence about observation of animal health. Please write some sentences on animal health, also in the results part.
R: Done. We have added the suggested information. Sorry, but we don't measure animal health parameters like parasites.
Line 184-186: Before the commencement of the trial, all animals were treated with an anthelmintic, and the control of external parasites such as ticks and horn-fly was performed when necessary (Molento et al., 2013; Dias de Castro et al., 2017).
- After Line 235: Could you please also mention at what age (days) the animals were slaughtered in both systems; at least the ones that were not added to the system HILF: in the discussion part you mention that it is more efficient (and also has an impact on meat qualtity and climate) to fatten animals faster, so it would be interesting to see this difference here, in the results part.
R: We do not slaughter the experimental animals. All animal variables measured were with live animals.
The mention of slaughter time and carcass quality are discussed in the manuscript mentioning the results of references Villalba and Provenza, 2009; Carvalho et al., 2006; and Savian et al., 2021 (studies similar to this manuscript). The idea of this paragraph was to present problems and opportunities regarding treatment outcomes.
- Chapter 4.: Discussion: I think your work is very interesting and important and I also like, how you discuss it. But I thought that some sentences on fertilizing (animal manure / plant residuals) and soil compaction should be added. Do you expect them to be about the same in both systems or are there differences or do you think such differences could be observed in further studies or on the long term? These subjects should at least be mentioned in the discussion part.
R: Ok. We add these considerations to the manuscript.
Regarding the effect of the animal's hoof on soil compaction, we understand that in a short-term experiment the effect is not detectable. For this, a long-term study would be necessary.
In long-term experiments, small compaction of soil was found when grazing pressure was high, however, at moderate grazing intensities this compaction was not detected (Flores et al., 2007). Therefore, we believe that in the LIHF there is no soil compaction by animals, whereas in the HILF treatment there may be superficial compaction, but it can only be detected in a long-term experimental design.
Flores JPC, Anghinoni I, Cassol LC, Carvalho PCF, Leite JGB, Fraga TI (2007) Soil physical attributes and soybean yield in an integrated livestock-crop system with different pasture heights in no-tillage. R. Bras. Ci. Solo 31, 771-780
Line 335-341: However, the herbage mass not harvested by the animals in each strip will remain to the next stocking cycle to be grazed, mainly because the rest period is short (high grazing frequency). Thus, the greater herbage production in the LIHF treatment can contribute to nutrient recycling and soil conservation in the long-term. For instance, Assmann et al. (2014) showed that heavy grazing intensity results in losses of carbon and nitrogen and soil organic matter degradation when compared with moderate and light grazing intensities.
Reviewer 2 Report
Dear authors,
I consider that you have done a good job, however, I would like to point out some errors and some details that require changes or clarifications. They are as follows:
- The design of the Methodology has some weaknesses:
- The sample is small, although this fact has been partially compensated by the number of repetitions.
- I have not seen that the effect of trampling cattle has been considered.
- I don't know how the selective behavior of cattle when consuming grass is controlled in the LIHF system.
- While in the Abstract, 16 steers weigh 290 kg, in the text, the steers weigh 265 kg.
- In Table 1, the “Leaf: stem ratio” is not correctly located.
- Sorghum has responded as expected, taking into account other similar research works carried out with grasses (cereals), which have been pointed out in the Discussion. However, most grasslands include legumes and other pasture plants, and I do not know if the results could be extrapolated to these other types of grasslands. I would like the authors to answer the following question: do you think that the LIHF system is also better for other types of grasslands?
Kind regards
Author Response
I consider that you have done a good job, however, I would like to point out some errors and some details that require changes or clarifications. They are as follows:
The design of the Methodology has some weaknesses:
- The sample is small, although this fact has been partially compensated by the number of repetitions.
R: We understand your concern about the sample size (8 experimental units). Although the number of experimental units is small, we had repeated measures in time to increase the residual degrees of freedom, ensuring a minimum of 10 degrees of freedom from error (Pimentel Gomes, 1978).
Pimentel Gomes, F. (1978). Curso de Estatística Experimental, 8a. Edição, Livraria Nobel, São Paulo.
- I have not seen that the effect of trampling cattle has been considered.
R: You are right to be concerned about cattle trampling on soil and vegetation.
For the effect of trampling on vegetation, we found that this was most important at the beginning of the experiment when the plants were more tender and had not been grazed. In this case, we were careful not to collect the "trampled" forage mass and measured the canopy heights when they were lying down (living plant parts). However, over time, plants and animals adapt to management and the effect of trampling on vegetation is much less. However, we don’t measure that.
- I don't know how the selective behavior of cattle when consuming grass is controlled in the LIHF system.
R: We agree with reviewer 2. We do not believe that selective animal behavior is controlled in the LIHF treatment.
What we suggest is that in the LIHF system it is possible to have a minimum control of the sward structure in which the animal will consume. This is because we know that animals prefer leaves and in this management target (50 and 30 cm for pre- and post-grazing, respectively) high amounts of leaves are offered to the animals during grazing. Regarding intake selectivity, there is some evidence to suggest that in the LIHF system (more heterogeneous heights structure in the LIHF paddock in relation to HILF) animals can be more selective in consuming the plant parts that best meet the nutritional demands of animals.
For more information see:
Savian JV, Schons RMT, Mezzalira JC, Neto AB, Neto GDS, Benvenutti MA and Carvalho PCF (2020) A comparison of two rotational stocking strategies on the foraging behaviour and herbage intake by grazing sheep. Animal 14 (12), 2503–2510.
Savian JV, Schons RMT, de Souza Filho W, Zubieta AS, Kindlein L, Bindelle J, Bayer C, Bremm C and de Carvalho PCF (2021) ‘Rotatinuous’ stocking as a climate-smart grazing management strategy for sheep production. Science of the Total Environment 753, 141790.
Zubieta AS, Marín A, Savian JV, Soares Bolzan AM, Rossetto J, Barreto MT, Bindelle J, Bremm C, Quishpe LV, Valle SDF, Decruyenaere V and de F. Carvalho PC (2021a) Low-intensity, high-frequency grazing positively affects defoliating behavior, nutrient intake and blood indicators of nutrition and stress in sheep. Frontiers in Veterinary Science 8, 631820.
Zubieta ÁS, Savian JV, de Souza Filho W, Wallau MO, Gómez AM, Bindelle J, Bonnet OJF and Carvalho PCF (2021b) Does grazing management provide opportunities to mitigate methane emissions by ruminants in pastoral ecosystems? Science of The Total Environment 754, 142029.
- While in the Abstract, 16 steers weigh 290 kg, in the text, the steers weigh 265 kg.
R: Done. We've changed the text to the correct animal weight.
Line 35-40: The management target of 50 and 30 cm for pre- and post-grazing, respectively, a LIHF grazing management strategy oriented to maximize beef cattle herbage intake per unit time, was compared with a HILF grazing management strategy of 80 and 20 cm for pre- and post-grazing, respectively, aiming to maximize herbage accumulation and harvest efficiency. Sixteen Brangus steers of 15-month-old and 265 ± 21 kg of live weight (LW) were randomly distributed to pad-docks (experimental units).
- In Table 1, the “Leaf: stem ratio” is not correctly located.
R: Done. We modified the location in the table.
- Sorghum has responded as expected, taking into account other similar research works carried out with grasses (cereals), which have been pointed out in the Discussion. However, most grasslands include legumes and other pasture plants, and I do not know if the results could be extrapolated to these other types of grasslands. I would like the authors to answer the following question: do you think that the LIHF system is also better for other types of grasslands?
R: Yes! we know that for Italian ryegrass the results (for HILF and LIHF) are very similar to this manuscript. We believe that for pastures with a habit-growing similar to sorghum and Italian ryegrass the result should have the same logic.
However, for perennial and/or habit-growing species (and response functional) different, we do not know what the result will be. It would be very interesting to study the response of HILF and LIHF with these different types of plant species.
Reviewer 3 Report
The aim of the study was to evaluate the effects of high-intensity and low-frequency vs low-intensity and high-frequency (LIHF) grazing strategies on herbage production and performance of beef cattle grazing sorghum pastures.
The study is original and very interesting given the importance of identifying grazing systems that can, at the same time, reduce the environmental impact of animals on the grazing resource without compromising their production performance.
All sections of the paper are well written.
Just a small suggestion: in material and methods (L. 172-174) I would replace the within text description of the stocking rate calculation with a formula that makes it easier to understand.
Author Response
Done. We added the formula in the manuscript that makes it easier to understand.